materials science/biomaterials/biophysics

troponin I, surface plasmon,
single monoclonal antibody, biosensor

**Authors for correspondence:**
Reham F. El-Kased
e-mail: amal.kasry@bue.edu.eg
Jakub Dostalek
e-mail: jakub.Dostalek@ait.ac.at
Amal Kasry
e-mail: reham.kased@bue.edu.eg

# Development of a specific troponin I detection system with enhanced immune sensitivity using a single monoclonal antibody

Anıl Bozdogan[1,2], Reham F. El-Kased[3],
Vanessa Jungbluth[2], Wolfgang Knoll[1,2], Jakub Dostalek[2]
and Amal Kasry[1,4]

[1]CEST Competence Centre for Electrochemical Surface Technology, 2700 Wiener Neustadt, Austria
[2]Biosensor Technologies, AIT-Austrian Institute of Technology GmbH, Konrad-Lorenz-Straße 24, 3430 Tulln, Austria
[3]Department of Microbiology and Immunology, Faculty of Pharmacy, and
[4]Nanotechnology Research Centre (NTRC), The British University in Egypt (BUE), El-Sherouk City, Suez Desert Road, Cairo 11837, Egypt

AK, 0000-0002-8130-8693

Using an immunoassay in combination with surface plasmon fluorescence spectroscopy (SPFS), we report the rapid detection of troponin I, a valuable biomarker for diagnosis of myocardial infarction. We discuss the implementation of (i) direct, (ii) sandwich, and (iii) competitive assay formats, based on surface plasmon resonance and SPFS. To elucidate the results, we relate the experiments to orientation-dependent interaction of troponin I epitopes with respective immunoglobulin G antibodies. A limit of detection (LoD) of 19 pM, with 45 min readout time, was achieved using single monoclonal antibody that is specific for one epitope. The borderline between normal people and patients is 20 pM to 83 pM cTnI concentration, and upon the outbreak of acute myocardial infraction it can raise to 2 nM and levels at 20 nM for 6–8 days, therefore the achieved LoD covers most of the clinically relevant range. In addition, this system allows for the detection of troponin I using a single specific monoclonal antibody, which is highly beneficial in case of detection in real samples, where the protein has a complex form leading to hidden epitopes, thus paving the way towards a system that can improve early-stage screening of heart attacks.

# 1. Introduction

Cardiovascular diseases (CVD) refer to several types of conditions that affect the heart and blood vessels, which include acute myocardial infarction (AMI), coronary heart disease (CHD), cerebrovascular disease, peripheral arterial disease, rheumatic heart disease and congenital heart disease. According to the World Health Organization (WHO), CVD is the leading cause of death worldwide, where it is responsible for 30% of all deaths globally and WHO estimates that by 2030, 23.6 million people will die from CVD [1]. Low- and middle-income countries often lack the integrated principal healthcare programmes for early diagnosis and treatment of people with risk factors compared with people in high-income countries, causing the annual costs of CVD management worldwide to be extremely high. In Egypt, as an example, WHO published in 2014 that CHD deaths reached 23.14% of total deaths, which ranks Egypt as the twenty-third in CHD worldwide [2]. Therefore, there is a strong demand for a robust and economical approach for early diagnosis and prognosis of CVD.

The use of classical CVD biomarkers, such as creatine kinase (CK) and lactate dehydrogenase, has been restricted owing to absence of tissue specificity and sensitivity. In 2000, the European Society of Cardiology and the American College of Cardiology recommended the use of troponin as a biomarker for the diagnosis of AMI [3], owing to the presence of tissue-specific isoforms in cardiac muscle. Troponin complex (cTn) is a heterogenic protein, which plays an important role in the regulation of cardiac muscle contraction. The troponin complex consists of three subunits: troponin T (cTnT), troponin I (cTnI) and troponin C (cTnC). Being expressed only in cardiac tissue, troponins I and T have been the preferred biomarkers for myocardial infarction for a long time [4]. cTnI is confined inside the heart muscle and it is, therefore, considered to be the standard biomarker for detecting AMI, because it is significantly more specific than other heart markers [5]. Early troponin I detection would lead to faster diagnosis and consequently the initiation of the correct treatment, which improves the prognosis for patients. It has been demonstrated that testing troponins on patient admission and again after 6–12 h provides better risk stratification and early diagnosis [6]. cTnI levels begin to rise 2–3 h after the myocardial infarction and elevation of its levels can persist for up to 10 days, making it ideal for retrospective diagnosis of infarctions [7–9]. The borderline between normal people and patients is 20 pM to 83 pM cTnI concentration, while after the outbreak of AMI, this concentration can go up to 2 nM within 3–6 h, and levels at about 20 nM for 6–8 days [10].

Several epitopes are identified on the cTnI protein; of which only six are used in laboratory research and clinical research (aa 24–40, aa 22–43, aa 41–49, aa 83–93, aa 87–91, and aa 186–192, respectively). Five of these epitopes are located on the N-terminus, which is considered the leading antigenic region of the recombinant version of cTnI protein (TNNI3 sequence) [11–13]. Different diagnostic assays have been designed for the quantitative measurements of cTnI in human blood using monoclonal antibodies, which have found widespread clinical applications as diagnostic and therapeutic agents for different diseases [14]. However, this approach lacks robustness and the common reason for the inconsistency between cTnI assay measurements can be attributed to the difference in the epitope specificity of the antibodies used in various assays. In addition, some epitopes are lost as a result of degradation of circulating troponin I, whereas others remain unaltered, resulting in different recoveries by different assays [15].

Surface plasmon resonance (SPR) biosensors represent an established technique for rapid detection and interaction analysis of biomolecules on solid surfaces [16–19]. In this technique, resonantly excited surface plasmons are used for probing the capture of specific target analytes from the analysed sample at the sensor surface. Surface plasmons originate from collective oscillations of the electron density in the metal coupled to an associated electromagnetic field that is confined to the metallic layer or metallic nanostructures deployed at the sensor surface. The analyte binding-induced refractive index change detunes the resonant excitation of surface plasmons and thus is converted to an optical signal. In addition, the resonant excitation of surface plasmons generates an enhanced electromagnetic field intensity, which can be employed for the amplification of weak spectroscopy signals such as scattering [20] and fluorescence [21]. Up to now, the probing with surface plasmons was exploited in a range of biosensors for rapid and sensitive detection of cardiac biomarkers including those based on SPR supported by metallic nanoparticles [22], gold nanoparticle aggregation assay with colorimetric readout [23], surface plasmon-enhanced scattering [20], and localized surface plasmon-enhanced fluorescence [24].

The detection of the full-length cTnI is usually performed either by polyclonal antibody or multiple monoclonal antibodies specific for several epitopes [9,25,26], in order to increase the sensitivity and the specificity, which is important in case if the protein is in its complex form.

This work reports, for the first time to our knowledge, on using surface plasmon-enhanced fluorescence spectroscopy (SPFS) assay for the sensitive detection of cTnI at clinically relevant

concentrations, using only one monoclonal antibody specific for one cTnI epitope. Moreover, it attempts to elucidate the role of the biointerface design by combined SPR and SPFS study in order to optimize the assay performance characteristics. This study is supported by protein modelling in order to visualize the dependence of the epitopes exposure on the protein orientation, which plays a significant role and it may provide important insights into the future development of cTnI immunoassays for accurate screening of CVD at early stage.

# 2. Material and methods

## 2.1. Materials

Recombinant cTnI protein TNNI3 and anti-(TNNI3) goat polyclonal antibody were purchased from MyBioSource, San Diego, CA, USA (cat. no. MBS2010502 and cat. no. MBS 833132, respectively). The cTnI protein was reconstituted in phosphate buffer saline (PBS), pH 7.4, to obtain a final concentration of 100 µg ml$^{-1}$. High performance chromatography was done to purify and remove any salts from the protein. Monoclonal mouse immunoglobulin G (IgG) against cTnI epitope aa 87–91, Goat anti-mouse IgG conjugated with Alexa Flour 647, and Donkey anti-goat IgG Alexa Flour 647, were all purchased from antibodies-online, USA.

## 2.2. Gel electrophoresis

1D sodium dodecyl sulfate-polyacrylamide gel electrophoresis (SDS-PAGE) was performed for the full-length recombinant cTnI to determine purity and molecular weight, and 13 5 µl of pure sample was mixed with 5 µl sample buffer [13] (156 mM (hydroxymethyl)aminomethane (TRIS), 5% SDS, 25% glycerol, 0.5% bromophenol blue and 12.5% 2-mercaptoethanol) [13]. The mixture (10 µl) was loaded on a Bolt 4–12% Bis-Tris Plus Gels, 1.0 mm × 15 well (Life Technologies Europe, Bleiswijk, the Netherlands). Gels were run at a constant voltage of 200 V for 50 min in 3-(N-morpholino)propanesulfonic acid (MOPS) buffer (0.025 M MOPS, 0.025 M TRIS, 3.465 mM SDS, 1.025 mM ethylenediaminetetraacetic acid (EDTA)) using the XCell SureLock Mini Cell electrophoresis chamber (Invitrogen, Karlsruhe, Germany). For all experiments, low range marker (MyBioSource, USA) was used [13]. Gels were stained using colloidal Coomassie Brillant Blue R-250 (MP, BIO, ICN, Santa Ana, CA, USA; cat. no. 821616).

## 2.3. Western blot

After performing the full-length protein to 1D SDS-PAGE as explained above, samples were blotted onto a polyvinylidene difluoride membrane [13] (Ji'an Qingfeng membrane Co., Ltd, Ji'an, China) by semi-dry blotting for 1 h with an electric current of 1.2 mA cm$^{-2}$, followed by cutting the membrane into strips and blocking with 2 ml buffer (Tris-buffered saline (TBS) 5% non-fat dry milk powder, 1% bovine serum albumin (BSA)) and incubation for 2 h [13]. The goat anti-cTnI polyclonal antibody (suspended in 0.01 M PBS [13], pH 7.4, concentration: 1 mg ml$^{-1}$) was used after 1 : 1800 dilution with blocking buffer [13]. Each strip was blocked and then incubated with 2 ml polyclonal antibody solution for 24 h at 4° C. This step was followed by three washing steps using 2 ml washing buffer per strip (TBS, 0.05% TWEEN 20 (v/v), 0.1% BSA (w/v)) [13]. The secondary antibody used for western blot was anti-goat IgG + HRP from MyBioSource cat. no. MBS 440120. The anti-goat IgG + horseradish peroxidase (HRP) (secondary antibody) solution was diluted to 1 : 20 000 with blocking buffer, 2 ml of this diluted solution was used to incubate the strip for 1 h at room temperature [13]. This was followed by a last washing step before visualization of the resulting band using CN/DAB substrate kit (Thermo Fisher Scientific Pierce Biotechnology, MA, USA, cat. no. 34000).

## 2.4. Sensor chip preparations and assay developing

A 50 nm gold layer was deposited on a high refractive index glass coated with 1.5 nm chromium, the gold layer was chemically modified with a mix of thiol with carboxy group and thiol with OH group with a ratio 1 : 9 (Sigma) by incubation overnight. The chips were then mounted on a sample holder while being attached to a flow cell of approximately 4 µl volume. The experiment was performed according to the following protocol: (i) the thiol layer was activated by a mix of 1-ethyl-3-(3-dimethylaminopropyl)carbodiimed/N-hydroxysuccinimide (EDC/NHS) (1 : 1 ratio) in order to

**Table 1.** List of the SPFS set-up devices.

| device | supplier |
| --- | --- |
| counter (53131A) | agilent (Santa Clara, USA) |
| fluorescence band pass filter (670FS10–25) | andover Corporation Optical Filter (Salem, USA) |
| laser notch filter (XNF-632.8–25.0 M) | CVI Melles Griot (Albuquerque, USA) |
| lock-in amplifier | EG&G (Gaithersburg, USA) |
| photomultiplier (H6240–01) | Hamamatsu Photonics (Hamamatsu, Japan) |

convert the COOH group to $NH_2$ group, this was done in situ using a pump at a flow rate (40 µl min$^{-1}$); (ii) cTnI protein was coupled to the sensor surface by the use of amine coupling; (iii) a solution with monoclonal mouse or polyclonal goat antibody (concentration of 1, 5 and 20 nM) was rinsed over this surface; followed by (iv) reaction of the affinity captured antibody with a secondary anti-mouse or anti-goat antibody, respectively. The secondary antibody was conjugated with Alexa Fluor 647 and dissolved in buffer at a concentration of 6 nM.

## 2.5. Surface plasmon resonance and surface plasmon-enhanced fluorescence spectroscopy measurements

An optical system combining SPR and SPFS was used for direct investigation of the interaction between the protein and its antibodies. The Kretschmann configuration, which is based on a high refractive index prism, 50 nm thermally evaporated gold glass slide, optically matched to the prism through matching oil, was used for the detection. A laser beam at wavelength of 633 nm was coupled to the prism and allowed to resonantly excite surface plasmons at the outer interface of thin gold film with a strength that was controlled by the angle of incidence. A flow cell with a volume of 4 µl was clamped against the gold sensor surface in order to flow liquid samples with a flow rate of 40 µl min$^{-1}$. The reflected beam intensity was measured with a lock-in amplifier in order to track changes in the SPR signal. The fluorescence signal measured on the SPFS modality was excited via the enhanced electromagnetic field intensity generated by surface plasmons. The emitted fluorescence light (at 670 nm) from the sensor surface was collected through the flow cell by a lens with a numerical aperture about NA = 0.2 and detected by a photomultiplier (H6240-01, Hamamatsu, Japan) that was connected to a counter. The intensity of the excitation beam irradiating the sensing area on the sensor chip of about 1 mm$^2$ was reduced to 30–60 µW to decrease bleaching of Alexa Fluor 647. The fluorescence light emitted at a wavelength of about 670 nm was spectrally separated from the excitation light at 633 nm by using a set of laser notch filter and fluorescence band pass filter (table 1).

## 2.6. Protein structure modelling

Protein structure modelling aims to determine spatial location of every atom in the protein molecule from the amino acid sequence by computational calculations. This was done using the I-TASSER server [27–29]. The server uses I-TASSER based algorithms to automatically generate high-quality model predictions for three-dimensional protein structure from their amino acid sequences. The I-TASSER detects structure templates from the Protein Data Bank using a fold recognition technique, which is based on protein modelling using known protein structures. The full-length structure models are constructed by reassembling structural fragments from the protein folding templates using replica exchange Monte Carlo simulations [30].

# 3. Results and discussion

Figure 1*a,b* shows SDS-PAGE of purchased recombinant full-length troponin cTnI to ensure its purity and molecular weight, confirming a molecular weight of 24 kDa. Figure 1*c* shows the Western blot results that confirm the affinity of the antibody to the full-length protein (more details are in the Material and methods section).

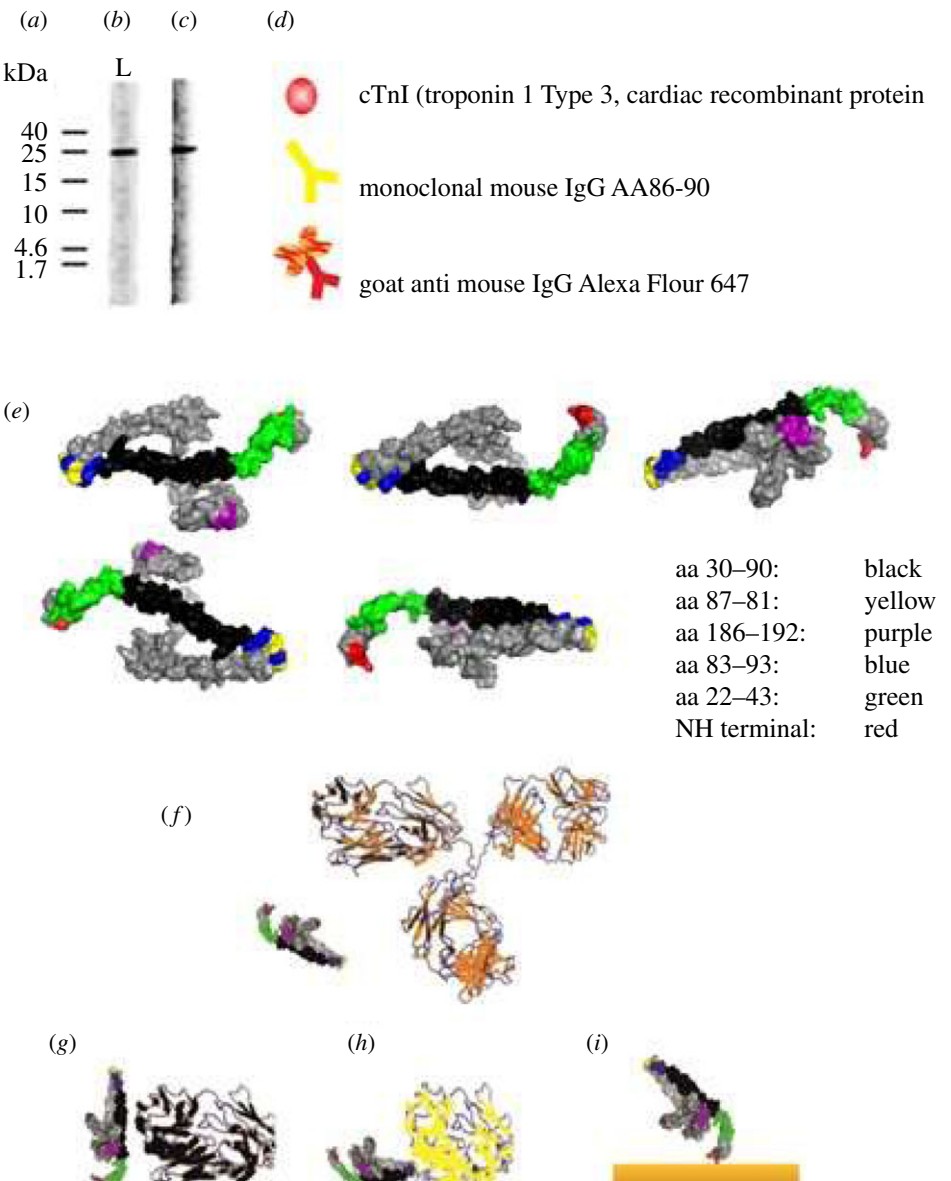

**Figure 1.** (a) cTnI protein marker lane (kDa), (b) 1D SDS–PAGE showing a single band at apparent molecular weight 24 kDa compared to the marker lane. (c) Western blot analysis against polyclonal anti-troponin antibody on a polyvinylidene fluoride (PVDF) membrane. (d) A schematic of the cTnI protein and the antibodies used in the experiments with their colours referring to their respective binding epitopes as shown in figure (e). (e) Modelled structure of full-length recombinant version of cTnI protein (aa 1–210) using I-TASSER software with its epitopes at different orientations. (f) The size comparison of immunoglobulin G antibody and cTnI ($\approx 1:3$). (g–i) shows the visualization of the possible cTnI protein orientation according to the binding epitope position; (g) possible orientation of cTnI protein that is affinity bound through its epitope aa 87–91 and (h) aa 30–90 to IgG Fab fragment. (i) The immobilization of the cTnI protein to the surface through its N-terminal (red) and the possible protein orientation.

Figure 1d provides an overview of the biomolecules that are further employed in the cTnI assay with plasmonic biosensor readout including the recombinant version cTnI protein and the respective antibodies (in colour respective to their binding protein epitopes). The structure of the full-length recombinant cTnI protein was visualized using bioinformatics methods for predicting the three-dimensional structure of protein molecules based on their amino acid sequences (more details are in the Methods section). Figure 1e shows the cTnI protein at different orientations where its epitopes are visualized with different colours, while the size difference between the protein and its antibody is compared in figure 1f. The same bioinformatics method was used to visualize the cTnI protein orientations in case of its affinity binding to mouse monoclonal antibody (recognizing cTnI epitope aa

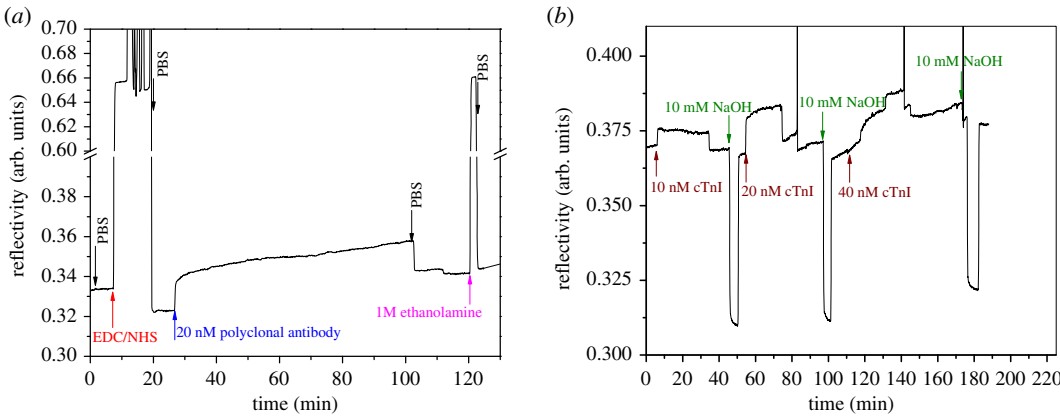

**Figure 2.** SPR observation of (*a*) immobilization of the polyclonal antibody on the surface modified with SAM, and (*b*) the subsequent affinity binding of the cTnI to the tethered polyclonal antibody with regeneration steps (10 mM NaOH applied) in between flow of cTnI samples.

87–91) and goat polyclonal antibody (recognizing cTnI epitope 30–90), or when coupled to the surface by using the N-terminus (figure 1*g,h,i*, respectively).

In order to bind both monoclonal and polyclonal antibodies to the cTnI protein, the possible protein orientations reveal that the epitope positions (aa 87–91 coloured in yellow) and (aa 30–90 coloured in black) spatially overlap, which may prohibit forming a sandwich.

Based on the Western blot results and the modelling, the cTnI detection was further investigated by using SPR and SPFS. On a first glance, we used the direct label-free SPR method for measuring the affinity binding of the cTnI to its polyclonal antibody immobilized on the sensor surface. To that end, we started with preparing a self-assembled monolayer (SAM), on the sensor gold surface, composed of thiols synthesized with oligo (ethylene glycol) (OEG) and carboxyl groups. Afterwards, the carboxyl groups were activated with a mix of EDC/NHS [31], followed by the amine coupling of the polyclonal antibody via its N-terminus or lysine groups figure 2*a*. As seen in figure 2*b*, the binding of cTnI showed measurable SPR signal only at a relatively high concentration of 20 nM. cTnI is known to have an isoelectric point in the range below 7 in case of AMI [32], which means that it would be negatively charged at the neutral pH that was used in the experiments. Because the gold surface is modified with a mix of thiol SAM carrying with carboxyl group and OEG groups, at pH 7, the surface would be negatively charged leading to repellence of the negatively charged protein, which in turn should prevent nonspecific binding. Although this charge effect is not strong enough to repel all negatively charged proteins, it is enough to avoid nonspecific binding. As the binding of the cTnI was not recognized until very high concentration, this was not further investigated, and a surface plasmon fluorescence-based protocol was developed.

In order to further confirm that the used monoclonal and polyclonal antibodies recognize the cTnI protein and to improve the sensitivity, SPFS readout modality was employed with the use of a secondary antibody conjugated to Alexa Fluor 647 label. As seen in figure 3*a*, the affinity binding of the antibody at the sensor surface does not change the fluorescence signal kinetics; however, the affinity binding of the respective secondary antibody leads to a strong change in the fluorescence signal $F(t)$. First, a rapid jump in fluorescence signal occurs owing to the excitation of Alexa Fluor 647-labelled molecules in the bulk, followed by gradual slower increase that is ascribed to the affinity binding of secondary antibody at the sensor surface. Upon the subsequent rinsing step, rapid decrease in the fluorescence signal is observed owing to replacing the solution with fluorophore-labelled molecules from the flow cell and then a slow decay in the signal is attributed to the dissociation of the attached biomolecules and bleaching of the fluorophore labels at the sensor surface. The fluorescence sensor response $\Delta F$ that is associated with the affinity binding (defined as the difference between the baseline signal before the injection of the antibody and the secondary antibodies and after the final rinsing) is linearly increasing with the concentration of the antibody. The unspecific interaction of the secondary antibody with the sensor surface (without captured anti-cTnI antibody) was tested and a negligible signal of $\Delta F = 132$ cps was measured for the anti-mouse secondary antibody (specific for the monoclonal antibody), however, a strong response of $\Delta F = 2790$ cps was observed for the anti-goat secondary antibody (specific for the polyclonal antibody), as shown in figure 3*b*. As was expected, the SPFS readout using a sandwich assay was not sensitive enough for low concentrations, owing to the close proximity of the epitopes aa 87–91 and aa 30–90

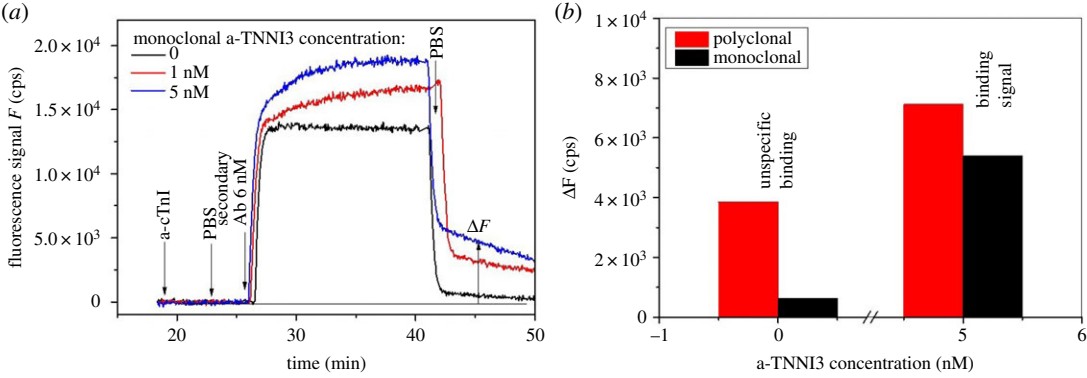

**Figure 3.** (a) Recognition of the monoclonal antibody to the cTnI protein, the black curve represents the control test, where 6 nM Alexa fluor 647 conjugated secondary antibody specific to the monoclonal cTnI was flown, no signal was observed after washing the surface. This was followed by binding of the secondary antibody to different concentrations of the monoclonal antibody bound to the cTnI, which gives clear fluorescence signal (red, blue) indicating successful binding. (b) A nonspecific binding test performed by flowing 6 nM Alexa fluor 647 conjugated two secondary antibodies, specific for both the monoclonal and the polyclonal antibodies, clearly showing that the unspecific interaction in case of the polyclonal antibody is higher than that of the monoclonal one (ΔF refers to fluorescence counts).

(figure 1), therefore an inhibition-competitive assay format was developed for the detection of cTnI analyte using the monoclonal anti-cTnI antibody. Although polyclonal antibodies, containing accurately identified antigenic determinants can provide an alternative to monoclonal antibodies for the detection of diseases biomarkers, whenever multipoint binding of the antigen is offered, which can result in an increased sensitivity of the assay owing to the increase in the avidity of antibody-antigen interaction [13], monoclonal antibodies are more advantageous than polyclonal ones. Their specificity is the most important advantage, as they only react with one epitope on the target molecule, while polyclonals contain hundreds or even thousands of specificities that can lead to cross-reactions and consequently much less selectivity. In addition to that, our SPFS results clearly indicate that the nonspecific binding in case of the polyclonal antibody is much higher (figure 3b), therefore we chose to develop the competitive assay using the monoclonal antibody. Given that cTnI is released into the blood circulation of patients of AMI predominantly in its complex form, it is crucial to use antibodies which can recognize, not only free cTnI, but also cTnI complexed with other cTn subunits [33], monoloclonals would be more advantageous in that regard. It is worth mentioning that the chosen monoclonal mouse IgG against cTnI epitope aa 87–91 is specific for cTnI where the epitope (87–91; GLGFA) is not found through the whole troponin T sequence, therefore, it cannot cross-react with cTnT and it can distinguish between both of them [34], electronic supplementary material.

As schematically shown in figure 4a, the developed cTnI assay with SPFS readout consisted of three steps. First, a sensor chip was prepared with cTnI protein immobilized to a mixed thiol SAM by amine coupling. Second, the analysed sample with cTnI was spiked with mouse monoclonal antibody at a concentration of 2 nM and reacted for 20 min. Afterwards, the sample was flowed over the sensor surface for 20 min followed by rinsing for 5 min. Third, a secondary anti-mouse antibody solution (6 nM) was reacted for 20 min with the affinity bound antibody at the sensor surface. After rinsing for 5 min, the fluorescence sensor response ΔF was determined. The established calibration curve in figure 4b shows that the sensor response ΔF is inversely proportional to the target cTnI analyte concentration. In a sample with low amounts of cTnI analyte, the majority of binding sites of the antibody spiked into the sample are free and can react with cTnI coupled to the sensor surface. Therefore, a large fluorescence signal is generated after the reaction with the secondary antibody. For large amounts of cTnI target analyte molecules present in the sample, the binding sites of the antibody are occupied, and do not bind to the surface, thus leading to a low fluorescence signal response. In order to calculate the limit of detection (LoD), the response of the fluorescence signals was plotted as a function of the cTnI analyte concentrations (figure 4c), and the curve was fitted using the sigmoidal function [35,36]:

$$f(x) = \frac{(A_1 - A_2)}{1 + (c/c_0)^p} + A_2.$$

The achieved LoD is 19 pM, it was determined as the concentration for which the fitted calibration curve intersects the fluorescence signal measured for a blank sample (ΔF0 = 132 cps) lowered by three times the standard deviation of noise (σ = 10.6 cps). Although this achieved LoD is not the lowest

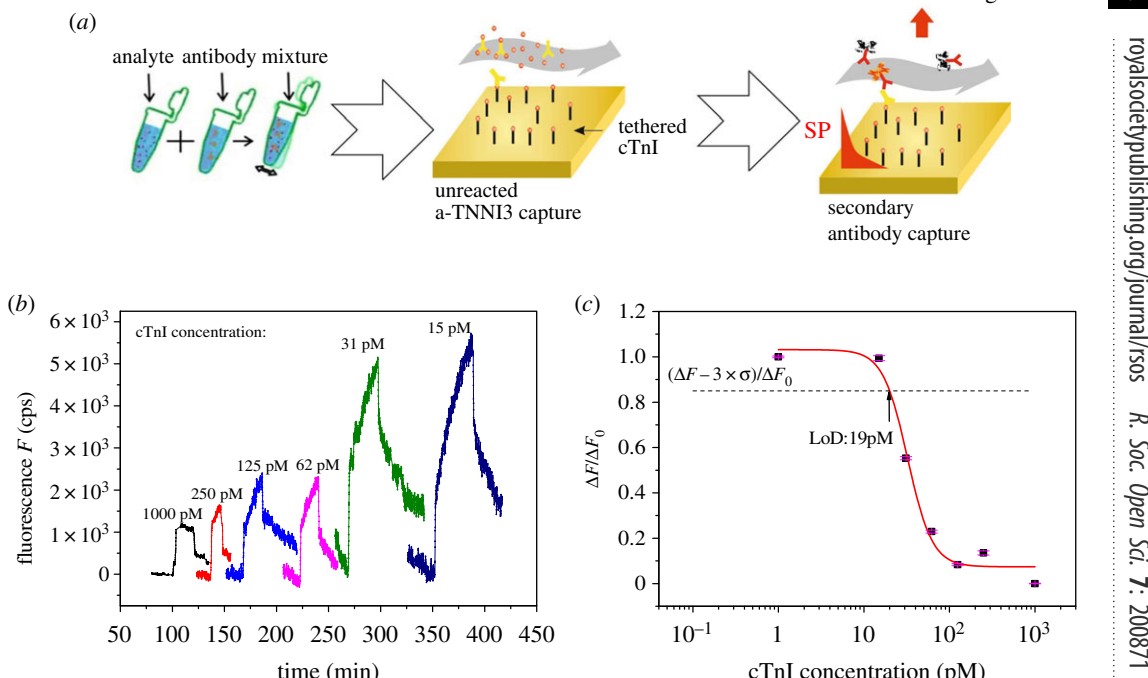

**Figure 4.** (*a*) A schematic of the inhibition-competitive assay steps, (*b*) fluorescence signal kinetics measured upon the binding of the secondary antibody conjugated with Alexa Fluor 647 and varied concentrations of cTnI, and (*c*) normalized calibration curve of the SPFS biosensor with an indicated LoD of 19 pM.

among the previously investigated detection methods of cTnI, it is to the authors' knowledge, the first time to detect this protein using a single monoclonal antibody specific for one epitope, which would be very beneficial to use in case of using real samples where the protein is known to be in its complex form. Detailed summaries of the other detection methods and LoDs are in [10,37].

It is worth noting that the proposed SPFS readout is inherently more tolerant to fouling of the sensor surface that inevitably occurs on the gold sensor surface with thiol SAM carrying oligo(ethylene glycol) chains when realistic samples such as blood serum [38] are tested. For the analysis of biomarkers in more complex samples including saliva and blood plasma, more advanced biointerfaces based on anti-fouling polymer brushes [39,40] can be deployed on the surface of plasmonic biosensors as was previously reported by our laboratory. In combination with the advancement of SPFS instrumentation and biointerfaces (which were not the subject of the present research reported here), the reported results make an important step towards developing affinity biosensors that can serve for rapid diagnosis of cardiac diseases at an early stage with reliable and reproducible results.

# 4. Conclusion

We have developed a sensitive troponin cTnI immunoassay, using a single monoclonal antibody, with an LoD of 19 pM in 45 min detection time. It takes advantage of SPFS detection by optically probing the enhanced field intensity at the fluorophore label absorption wavelength ($\lambda_{abs}$: 633 nm). We illustrate that the direct detection format based on the regular SPR biosensor principle does not provide sufficient sensitivity for the analysis of cTnI at clinically relevant concentrations. By using the same instrument operated in SPFS modality, we significantly improved the sensitivity by implementing an inhibition-competitive immunoassay. Crucial for this assay format is the careful selection of cTnI epitope and respective monoclonal antibody based on both experimental trials and theoretical modelling. These studies confirmed that immobilizing the ligand (recombinant version of target cTnI protein) directly to the surface via its N-terminal leaves the epitopes aa 30–90 and aa 87–91 available for affinity binding of respective IgG antibodies.

Data accessibility. All the necessary data are included in the main manuscript and figures, and raw data for all the figures in the paper are uploaded as electronic supplementary material. BLAST results are presented in the electronic supplementary material file.

Authors' contributions. R.F.E. performed the SDS–PAGE, Western blot analysis and the protein modelling, A.B., V.J. and A.K. designed and performed the SPR experiments. A.K., A.B and R.F.E. wrote the paper, J.D. and W.K. contributed to both the design of the experiments and writing the paper. All authors contributed to different aspects of the work, including writing and reviewing the paper.

Competing interests. We declare we have no competing interests.

Funding. The authors would like to thank the Competence Centre for Electrochemical Surface Technology (CEST) and the British University in Egypt (BUE) for financial support.

Acknowledgements. J.D. acknowledges support from the European Union's Horizon 2020 research and innovation program under the Marie Sklodowska-Curie grant agreement no. 64268.

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
