## [Reviewer comments · Royal Society Open Science]

Review History

RSOS-200871.R0 (Original submission)

Review form: Reviewer 1

Is the manuscript scientifically sound in its present form?

Yes

Are the interpretations and conclusions justified by the results?

Yes

Is the language acceptable?

Yes

Do you have any ethical concerns with this paper?

No

Have you any concerns about statistical analyses in this paper?

No

Recommendation?

Accept as is

Comments to the Author(s)

The corrected version of the manuscript is acceptable for publication.

Review form: Reviewer 2

Is the manuscript scientifically sound in its present form?

Yes

Are the interpretations and conclusions justified by the results?

Yes

Is the language acceptable?

Yes

Do you have any ethical concerns with this paper?

No

Have you any concerns about statistical analyses in this paper?

No

Recommendation?

Accept with minor revision (please list in comments)

Comments to the Author(s)

Please, find the comments in the attached file (Appendix A).

Decision letter (RSOS-200871.R0)

Dear Dr Kasry

On behalf of the Editors, we are pleased to inform you that your Manuscript RSOS-200871 "Development of a Specific Troponin I Detection System with Enhanced Immune Sensitivity using a Single Monoclonal Antibody" has been accepted for publication in Royal Society Open Science subject to minor revision in accordance with the referees' reports. Please find the referees' comments along with any feedback from the Editors below my signature.

Please submit your revised manuscript and required files (see below) no later than 7 days from today's (ie 01-Sep-2020) date. Note: the ScholarOne system will 'lock' if submission of the revision

is attempted 7 or more days after the deadline. If you do not think you will be able to meet this deadline please contact the editorial office immediately.

on behalf of Pietro Cicuta (Subject Editor)
openscience@royalsociety.org

Associate Editor Comments to Author:

Comments to the Author:

Thank you for engaging so constructively with the concerns of the reviewers following transfer from JRSI. A number of comments remain to be tackled.

Reviewer comments to Author:

Reviewer: 1

Comments to the Author(s)

The corrected version of the manuscript is acceptable for publication.

Reviewer: 2

Comments to the Author(s)

Please, find the comments in the attached file.

===PREPARING YOUR MANUSCRIPT===

Please ensure that you include an acknowledgements' section before your reference list/bibliography. This should acknowledge anyone who assisted with your work, but does not

qualify as an author per the guidelines at <https://royalsociety.org/journals/ethics-policies/openness/>.

===PREPARING YOUR REVISION IN SCHOLARONE===

<https://royalsociety.org/journals/authors/author-guidelines/#data>. You should ensure that

you cite the dataset in your reference list. If you have deposited data etc in the Dryad repository, please only include the 'For publication' link at this stage. You should remove the 'For review' link.

Author's Response to Decision Letter for (RSOS-200871.R0)

See Appendix B.

Decision letter (RSOS-200871.R1)

Dear Dr Kasry,

It is a pleasure to accept your manuscript entitled "Development of a Specific Troponin I Detection System with Enhanced Immune Sensitivity using a Single Monoclonal Antibody" in its current form for publication in Royal Society Open Science. The comments of the reviewer(s) who reviewed your manuscript are included at the foot of this letter.

on behalf of the Associate Editor and Pietro Cicuta (Subject Editor)
openscience@royalsociety.org

Associate Editor Comments to Author:

Thank you for submitting your revised manuscript and point-by-point response to the referees. I believe you have responded sufficiently to the referees' concerns and that your manuscript can now be accepted as is. Thank you for choosing Royal Society Open Science for your study.

Appendix A

Comments:

The authors employed Surface Plasma Resonance (SPR) sensing platform for specific detection of the Troponin I based on the immunoassay including monoclonal antibody and single epitope-specific interaction, and also offered an enhanced signal derivation adding a fluorescence approach. The study is interesting, however needs some revisions to be suitable for publication.

The comments are as follows-

1. Introduction is well-written to highlight the research objective.
2. Page 5, line 20 : (2 of 'mA/cm²' should be superscripted)
3. Page 5, line 20 : 'suspend-ed' should be revised to 'suspended'
4. Page 5, line 27-28 : Please, consider revision of the following sentence for clear understanding –
'The anti-goat IgG + HRP secondary antibody solution was diluted 1:20,000 with blocking buffer, then added, 2 ml per strip, and incubated for 1 h at room temperature [13].'
5. Page 5, line 35 and line 45 : '4 @L' need to be revised
6. Page 5, line 41-42: Please, consider revision of the following sentence for better understanding
'The Kretschmann configuration was used for the detection; a sensor chip with 50 nm gold layer on the top was optically matched to the base of high refractive index glass prism.'
7. Page 5, line 52: ...of about 1 mm² was reduced... need to revise (2 of '1 mm²' should be superscripted)
8. Page 6, line 48-49: '.....as shown in e, (e) Modeled.....' can be revised to '.....as shown in figure (e), (e) Modeled.....'
9. Page 7, line 3 from the top: 'Fig1c' instead of 'Figure 1(c)' ... Please, maintain the similar style throughout texts.
10. Page 7, line 16: '..... the N-terminus, Figure 1 (g, h, and i, respectively).....' can be revised to '..... the N-terminus [Figure 1 (g-i)].....'
11. Page 8, line 18-21 (caption of Figure 3(a)) :
 - (i) Please, consider revision. If possible, split it in several short sentences.
 - (ii) Please, clarify the statement 'no signal was observed after washing the surface (black)' whether it is for blank or control.

(iii) at line 18 and 22 : '..... 6 nM labeled ' could be '..... 6 nM fluorescent-labeled '. Please, verify. ('fluorescent' can be the name of dye)

12. Page 8, Figure 3 (b): How many times, the experiment was repeated? Probably, standard error bar is required.

13. Page 8, line 31-35:

(i) "The experiment at a concentration of 6 nM." This part is suitable for experimental (materials and methods) section. Please, consider to add this part to the section "3.4 Sensor chip preparations" and can rename the heading as "Sensor chip preparations and assay operations". Add relevant connecting sentences if necessary.

(ii) "The secondary antibody was conjugated with Alexa Fluor 647 and dissolved at a concentration of 6 nM." -----Dissolved to what? Is it OK to mention only "....conjugated with 6 nM Alexa Fluor 647"? If so, also revise other relevant sections (if applicable)

14. Page 8, line 53: '(Figure 1)' could be '[Figure 1]'. Please, follow similar style throughout the text.

15. Page 9, line 35-36: Please, consider revision for "and consequently much less sensitivity." for easy understanding. Also, please follow the journal's expression style regarding the words "cross reactions" or "cross-reactions"

16. Page 9, line 38-40: Please, consider revision for -

"Given that cTnI is released into the circulation of patients of AMI predominantly in its complex form, ..."

17. Page 9, line 43: Please, verify the expression style of the words "cross react" or "cross-react" and maintain the similar style throughout the text.

18. Page 9, line 45-52:

The part of "As schematically shown in Figure 4 (a), the developed cTnI assay ΔF was determined." is suitable for experimental section. Please, consider revision as described above for another similar case (of assay operations). Also, consider to move the schematic figure (i.e. Figure 4 (a)) to the site of corresponding text reflecting the principle of assay. If you do so, you may need to modify the Figure 4 keeping only the graphs.

19. Page 9, Figure 4(c): The X-Axis title should bear the unit in “()”. “cTnI concentration [pM]” should be “cTnI concentration (pM)”. The style should consistent throughout the texts.

20. Please, keep similarity in expressing the unit of microliter i.e. either ‘μl’ or ‘μL’ throughout texts.

21. Why, the 6 nM concentration of fluorescent dye was chosen for labelling? Need to be clarified.

22. Fluorescent-based plasmon enhancement is not new (Plasmonics (2014) 9:781–799, DOI 10.1007/s11468-013-9660-5). In which point, your technique (SPFS) is novel. Need to be explained.

23. The conclusion should be more straight-forward. Please, consider revision indicating the future application. Also consider the revision of the sentence “It takes advantage absorption wavelength.” Page 10, LINE 28-30

Appendix B

Amal Kasry
Nanotechnology Research Centre
(NTRC)
The British University in Egypt (BUE)
M: +20 101 6077575
Email: Amal.Kasry@bue.edu.eg

Anita Kristiansen
Editorial Coordinator

Dear Ms Anita Kristiansen,

We are extremely pleased to know that our manuscript: "*Development of a Specific Troponin I Detection System with Enhanced Immune Sensitivity using a Single Monoclonal Antibody*" by Anil Bozdogan, Reham F. El-Kased, Vanessa Jungbluth, Wolfgang Knoll, Jakub Dostalek, and Amal Kasry, has been accepted for publication in *the Journal of Royal Society Open Science*.

We here respond to the reviewer's comments, and we have modified the manuscript taking all the comments in consideration. We believe that now it is ready for publication and we are looking forward to see the article online soon.

Responses to the Reviewers' Comments on Manuscript ID RSOS-200871

Title: Detection of Cardiac Biomarker Troponin (cTnI) with Enhanced Immune Sensitivity

Author(s): Anil Bozdogan, Reham F. El-Kased, Vanessa Jungbluth, Wolfgang Knoll, Jakub Dostalek, Amal Kasry

Response to Reviewer #1:

The authors employed Surface Plasma Resonance (SPR) sensing platform for specific detection of the Troponin I based on the immunoassay including monoclonal antibody and single epitope-specific interaction, and also offered an enhanced signal derivation adding a fluorescence approach. The study is interesting, however needs some revisions to be suitable for publication.

The comments are as follows-

1. Introduction is well-written to highlight the research objective.

We thank the reviewer for this comment

2. Page 5, line 20 : (2 of 'mA/cm²' should be superscripted).

Done

3. Page 5, line 20 : 'suspend-ed' should be revised to 'suspended'.

Done

4. Page 5, line 27-28 : Please, consider revision of the following sentence for clear understanding –'The anti-goat IgG + HRP secondary antibody solution was diluted 1:20,000 with blocking buffer, then added, 2 ml per strip, and incubated for 1 h at room temperature [13].

Done

5. Page 5, line 35 and line 45 : '4 @L' need to be revised.

Done

6. Page 5, line 41-42: Please, consider revision of the following sentence for better understanding'The Kretschmann configuration was used for the detection; a sensor chip with 50 nm gold layer on the top was optically matched to the base of high refractive index glass prism.'

Done

7. Page 5, line 52: ...of about 1 mm² was reduced... need to revise (2 of '1 mm²' should be superscripted).

Done

8. Page 6, line 48-49: '.....as shown in e, (e) Modeled.....' can be revised to '.....as shown in figure (e), (e) Modeled.....'

Done

9. Page 7, line 3 from the top: 'Fig1c' instead of 'Figure 1(c)' ... Please, maintain the similar style throughout texts.

Done

10. Page 7, line 16: '..... the N-terminus, Figure 1 (g, h, and i, respectively).....' can be revised to '..... the N-terminus [Figure 1 (g-i)].....'

Done

11. Page 8, line 18-21 (caption of Figure 3(a)) : (i) Please, consider revision. If possible, split it in several short sentences.

(ii) Please, clarify the statement 'no signal was observed after washing the surface (black)' whether it is for blank or control.

(iii) at line 18 and 22 : '..... 6 nM labeled ' could be '..... 6 nM fluorescent-labeled '. Please, verify. ('fluorescent' can be the name of dye).

Done, sentences have been rephrased to be clearer

12. Page 8, Figure 3 (b): How many times, the experiment was repeated? Probably, standard error bar is required.

We repeated all the experiments between 2 and 3 times, the nonspecific binding tests are a standard one, and are being regularly practiced in our labs for the development of several biosensors. In Figure 3b we show that using polyclonal antibody leads to higher nonspecific binding, and this was one reason why we use a monoclonal one.

13. Page 8, line 31-35:

(i) "The experiment at a concentration of 6 nM." This part is suitable for experimental (materials and methods) section. Please, consider to add this part to the section "3.4 Sensor chip preparations" and can rename the heading as "Sensor chip preparations and assay operations". Add relevant connecting sentences if necessary.

(ii) "The secondary antibody was conjugated with Alexa Fluor 647 and dissolved at a concentration of 6 nM." -----Dissolved to what? Is it OK to mention only "...conjugated with 6 nM Alexa Fluor 647"? If so, also revise other relevant sections (if applicable).

Done, the part has been moved to the materials and methods section and sentences were rephrased.

14. Page 8, line 53: '(Figure 1)' could be '[Figure 1]'. Please, follow similar style throughout the text.

Done

15. Page 9, line 35-36: Please, consider revision for "and consequently much less sensitivity." for easy understanding. Also, please follow the journal's expression style regarding the words "cross reactions" or "cross-reactions"

Done

16. Page 9, line 38-40: Please, consider revision for -“Given that cTnI is released into the circulation of patients of AMI predominantly in its complex form, ...”

Done

17. Page 9, line 43: Please, verify the expression style of the words “cross react” or “cross-react” and maintain the similar style throughout the text.

Done

18. Page 9, line 45-52:

The part of “As schematically shown in Figure 4 (a), the developed cTnI assay ΔF was determined.” is suitable for experimental section. Please, consider revision as described above for another similar case (of assay operations). Also, consider to move the schematic figure (i.e. Figure 4 (a)) to the site of corresponding text reflecting the principle of assay. If you do so, you may need to modify the Figure 4 keeping only the graphs.

We thank the reviewer for the suggestion, we believe that this part is more suitable for the results section, as, although it describes a process, it represents the core of the idea which is developed based on the previous results. Also, to understand the results shown in Figure 4 b and c, the scheme is important. Therefore, we kept the figure and the description in the results section.

19. Page 9, Figure 4(c): The X-Axis title should bear the unit in “()”. “cTnI concentration [pM]” should be “cTnI concentration (pM)”. The style should consistent throughout the texts.

Done

20. Please, keep similarity in expressing the unit of microliter i.e. either ‘ μ l’ or ‘ μ L’ throughout texts.

Done

21. Why, the 6 nM concentration of fluorescent dye was chosen for labelling? Need to be clarified.

According to our experience in these type of experiments, 6nM is the lowest concentration that achieves signal saturation level, this is tested by creating a monolayer on the surface then we do a titration test to determine the lowest concentration that achieves saturation.

22. Fluorescent-based plasmon enhancement is not new (Plasmonics (2014) 9:781–799, DOI 10.1007/s11468-013-9660-5). In which point, your technique (SPFS) is novel. Need to be explained.

Indeed SPFS has been in several applications, however, our main aim in this work is to emphasize two things, the first is the detection using a single monoclonal antibody, which enhances the specificity, and the second is to show that the protein orientation plays an important role in the sensitivity, where the epitopes specific for the antibody should be exposed, this is the novelty in our work.

23. The conclusion should be more straight-forward. Please, consider revision indicating the future application. Also consider the revision of the sentence “It takes advantage absorption wavelength.” Page 10, LINE 28-30

Done